# Learn to Explain: Multimodal Reasoning via Thought Chains for Science Question Answering

**Pan Lu**[1,3], **Swaroop Mishra**[2,3], **Tony Xia**[1], **Liang Qiu**[1], **Kai-Wei Chang**[1],
**Song-Chun Zhu**[1], **Oyvind Tafjord**[3], **Peter Clark**[3], **Ashwin Kalyan**[3]
[1]University of California, Los Angeles, [2]Arizona State University, [3]Allen Institute for AI
{lupantech, kwchang.cs}@gmail.com, sczhu@stat.ucla.edu,
{oyvindt, peterc, ashwinkv}@allenai.org

## Abstract

When answering a question, humans utilize the information available across different modalities to synthesize a consistent and complete *chain of thought* (CoT). This process is normally a black box in the case of deep learning models like large-scale language models. Recently, science question benchmarks have been used to diagnose the multi-hop reasoning ability and interpretability of an AI system. However, existing datasets fail to provide annotations for the answers, or are restricted to the textual-only modality, small scales, and limited domain diversity. To this end, we present Science Question Answering (SCIENCEQA), a new benchmark that consists of ∼21k multimodal multiple choice questions with diverse science topics and annotations of their answers with corresponding lectures and explanations. We further design language models to learn to generate lectures and explanations as the *chain of thought* (CoT) to mimic the multi-hop reasoning process when answering SCIENCEQA questions. SCIENCEQA demonstrates the utility of CoT in language models, as CoT improves the question answering performance by 1.20% in few-shot GPT-3 and 3.99% in fine-tuned UnifiedQA. We also explore the upper bound for models to leverage explanations by feeding those in the input; we observe that it improves the few-shot performance of GPT-3 by 18.96%. Our analysis further shows that language models, similar to humans, benefit from explanations to learn from fewer data and achieve the same performance with just 40% of the data.[1]

## 1   Introduction

A long-standing goal of AI systems is to act reliably and learn complex tasks efficiently like human beings. In the process of reliable decision making, humans follow an explicit *chain-of-thought* (CoT) reasoning process that is typically expressed as an explanation. However, machine learning models are trained mostly using a large number of input-output examples to perform a specific task. These black-box models only generate the final decision without reliably revealing the underlying reasoning process. Not surprisingly, it is unclear if they understand the task and can generalize even though they perform well on the benchmark. On the other hand, humans are able to learn from instructions or explanations from past experience and generalize them to novel and unseen problems. This helps them learn more quickly with fewer data. In this work, we explore if machines can be endowed with such reasoning abilities in the context of science-based question answering.

Recently, science problem solving benchmarks [18] have been used to diagnose the multi-hop reasoning ability and interpretability of AI systems. To answer science questions, a model needs to

---

[1]The data and code are available at https://scienceqa.github.io.
 Work was partially done while Pan Lu and Swaroop Mishra were interns at AI2.

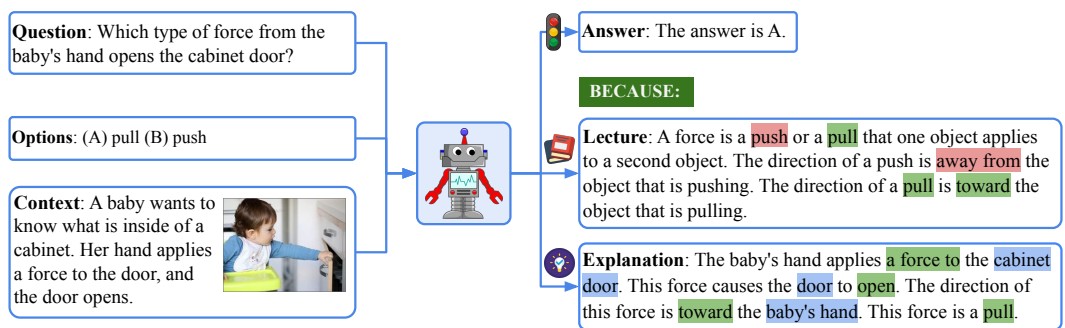

Figure 1: We construct the SCIENCEQA dataset where a data example consists of multimodal question answering information and the grounded lecture and explanation. We study if QA models can generate a reasonable explanation to reveal the chain-of-thought reasoning.

not only understand multimodal contents but also extract external knowledge to arrive at the correct answer. Since these tasks require domain-specific knowledge and explicit multi-hop reasoning, a model would be not interpretable if it fails to provide explanations to reveal the reasoning process. However, current science question datasets [18, 17, 52] mostly lack annotated explanations for the answers. To address this issue, other science datasets annotate the explanations, but they are restricted to the textual only modality and limited to small data scales [13, 6, 37] or a small set of topics [20, 14]. Therefore, we collect Science Question Answering (SCIENCEQA), a large-scale multi-choice dataset that contains multimodal science questions with explanations and features rich domain diversity.

SCIENCEQA is collected from elementary and high school science curricula, and contains 21,208 examples along with lectures and explanations. Different from existing datasets [17, 18, 52], SCIENCEQA has richer domain diversity from three different subjects: natural science, social science, and language science. A typical example consists of a question, multiple choices, multimodal contexts, a correct answer, as well as a lecture and an explanation. The lecture and explanation provide general external knowledge and specific reasons, respectively, for arriving at the correct answer.

Consider the thoughts one person might have when answering the question in Figure 1. One first recalls the knowledge regarding the definition of a force learned from textbooks: "*A force is a push or a pull that ... The direction of a **push** is ... The direction of a **pull** is ...*", then forms a line of reasoning: "*The baby's **hand** applies a force to the cabinet **door**. → This force causes the **door to open**. → The direction of this force is **toward** the baby's **hand**.*", and finally arrives at the correct answer: "*This force is a **pull**.*". Following [41], we formulate the task to output a natural explanation alongside the predicted answer. In this paper, we train language models to generate lectures and explanations as the *chain of thought* (CoT) to mimic the multi-hop reasoning process to answer SCIENCEQA questions.

Our experiments show that current multimodal methods [55, 1, 21, 9, 26, 35] fail to achieve satisfactory performance on SCIENCEQA and do not generate correct explanations. Instead, we find that CoT can help large language models not only in the few-shot learning setting but also in the fine-tuning setting. When combined with CoT to generate the lecture and explanation, the fine-tuned UnifiedQA [19] achieves an improvement of 3.99% as opposed to not using CoT in the fine-tuning stage. The few-shot GPT-3 model [4] via chain-of-thought prompting can obtain 75.17% on SCIENCEQA with an improvement of 1.20% compared to the few-shot GPT-3 without CoT. Prompted with CoT, GPT-3 can generate reasonable explanations as evaluated by automated metrics, and promisingly, 65.2% of explanations meet the gold standard of human evaluations. We also investigate the upper bound for models to harness explanations by including them in the input. We find that doing so improves GPT-3's few-shot performance by 18.96%, suggesting that explanations do aid models and are currently underutilized in the CoT framework. Further analysis shows that, like humans, language models benefit from explanations to learn with less data: UnifiedQA with CoT obtains the same results as UnifiedQA without CoT with only 40% of the training data.

To sum up, our contributions are three-fold: (a) To bridge the gap in existing datasets in the scientific domain, we build Science Question Answering (SCIENCEQA), a new dataset containing 21,208 multimodal science questions with rich domain diversity. To the best of our knowledge, SCIENCEQA is the first large-scale multimodal dataset that annotates lectures and explanations for the answers.

(b) We show that CoT benefits large language models in both few-shot and fine-tuning learning by improving model performance and reliability via generating explanations. (c) We further explore the upper bound of GPT-3 and show that CoT helps language models learn from fewer data.

## 2 Related Work

**Visual question answering.** Since the task of visual question answering (VQA) was first proposed in [2], there have been plenty of VQA datasets [56, 58, 23, 11, 15, 12] conducted to facilitate the research work. Although our SCIENCEQA dataset shares some features with VQA, there are several main differences between them. First, SCIENCEQA is more challenging than existing VQA datasets because it contains multimodal contexts and diverse topics in the scientific domain. In addition, most answers are annotated with lectures and explanations, which makes SCIENCEQA a suitable dataset for multi-modal question answering and multi-hop reasoning for AI systems. Inspired by the recent remarkable performance achieved for VQA [33, 32, 10, 9, 26, 7, 8], in this paper, we further extensively benchmark SCIENCEQA with a wide range of attention-based [1, 33, 21, 9] and Transformer-based [30, 26, 27, 7] methods.

**Datasets for science problems.** Science problem solving is a challenging task that requires an AI system not only to understand the multimodal information from the science curriculum but also to reason about how to answer the domain-specific questions. Current science problem datasets such as AI2D [17], DVQA [16], VLQA [52], and FOODWEDS [24] have contributed to multimodal reasoning in the scientific domain. For example, a portion of VLQA contains multimodal questions on science subjects. These datasets, however, lack annotated explanations for the answers to reveal the reasoning steps. Some other datasets annotate the answers in the forms of supporting facts [37, 20], entailment trees [6], explanation graphs [13], reasoning chains [14]. However, these datasets are restricted to the single text modality with small data scales and limited topics. Instead, our SCIENCEQA annotates the answers with grounded lectures and explanations. Besides, SCIENCEQA features a richer domain diversity across 3 subjects, 26 topics, 127 categories, and 379 skills.

**Learning from explanations and few-shot learning.** Explanations help humans understand a task better, and there have been several attempts to show the same for models. For example, the learning from instruction paradigm [40, 43, 53, 39, 45, 25], where the task level explanation is provided in the form of instruction, improves model performance significantly. An example of learning from explanations in the scientific domain is proposed in [51] where the model interprets demonstrative solutions to solve geometry problems. Recently, there has been a surge of interest in few-shot learning, where language models learn a specific task from a few examples [46, 3]. For instance, [42, 54, 34] find that explanations in the format of the chain of thought can improve language models' reasoning ability in few-shot learning. In this paper, we show that the chain of thought boosts the performance of large language models like UnifiedQA [19] if the models generate explanations along with the answer in a fine-tuning way. Furthermore, a few-shot GPT-3 model via chain-of-thought prompting is able to improve the reasoning performance on SCIENCEQA and generate reasonable explanations.

## 3 Dataset

We collect SCIENCEQA, which is a multimodal multiple-choice science question dataset containing 21,208 examples. An example in SCIENCEQA is shown in Figure 1. Given the science question and multimodal contexts, the task is to select the correct answer from multiple options. Different from existing datasets [50, 17, 52, 31, 24], SCIENCEQA covers diverse topics across three subjects: natural science, social science, and language science. Moreover, most questions are annotated with grounded lectures and detailed explanations. The lecture provides general knowledge that introduces the background information for solving problems of a similar class. The explanation reveals a specific reason for the answer. To effectively answer the questions, a model often needs to be able to understand the multimodal content in the input and extract external knowledge, similar to how humans do. More importantly, the goal of SCIENCEQA is to aid development of a reliable model that is capable of generating a coherent chain of thought when arriving at the correct answer to reveal the multi-step reasoning process. For data collection details, see Appendix A.1.

| Statistic | Number |
|---|---|
| Total questions | 21,208 |
| Questions with text context | 10,220 (48.2%) |
| Questions with image context | 10,332 (48.7%) |
| * Image of natural format | ≈2,960 (14.0%) |
| * Image of diagram format | ≈7,372 (34.8%) |
| Questions with both contexts | 6,532 (30.8%) |
| Questions without any context | 7,188 (33.9%) |
| Questions with a lecture | 17,798 (83.9%) |
| Questions with a explanation | 19,202 (90.5%) |
| Different questions | 9,122 |
| Different lectures | 261 |
| Topic classes | 26 |
| Category classes | 127 |
| Skill classes | 379 |
| Average question length | 12.11 |
| Average choice length | 4.40 |
| Average lecture length | 125.06 |
| Average explanation length | 47.66 |

Table 1: Main statistics in SCIENCEQA.

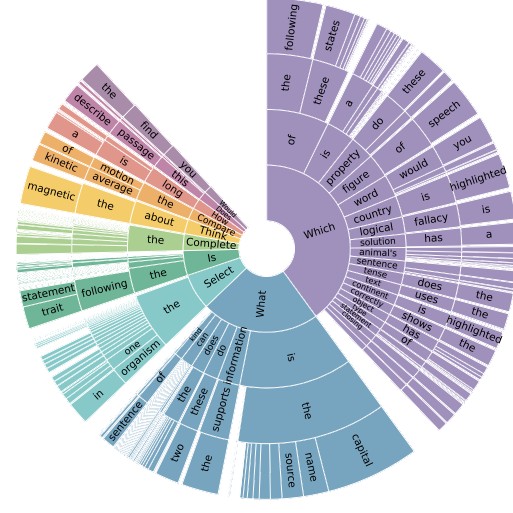

Figure 2: Question distribution in SCIENCEQA.

## 3.1 Data Analysis

**Key statistics.** We randomly split the dataset into training, validation, and test splits with a ratio of 60:20:20. Each split has 12,726, 4,241, and 4,241 examples, respectively. Table 1 shows the main statistics of SCIENCEQA. SCIENCEQA has a large set of different questions, totaling up to 9,122. Out of the 21,208 questions in SCIENCEQA, 10,332 (48.7%) have an image context, 10,220 (48.2%) have a text context, and 6,532 (30.8%) have both. 83.9% of the questions are annotated with a lecture, while 90.5% of the questions feature an explanation. The cross-combination of these information sources diversifies the problem scenario: sometimes the model is given a lot of information from multiple sources, while at other times, the only source of information is the question itself. This level of complexity is very common in grade-level science exams.

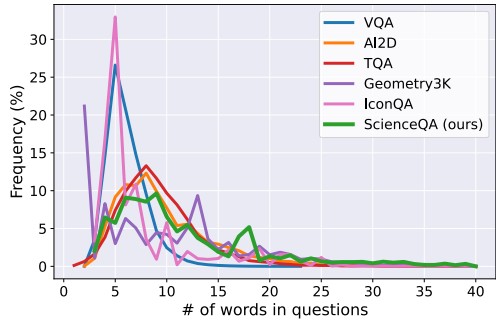

(a) Question length distribution of related datasets. SCI-ENCEQA is distributed more evenly in terms of the number of question words than other datasets.

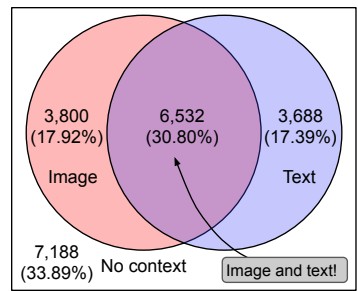

(b) Question distribution with different context formats. 66.11% of the questions in SCI-ENCEQA have either an image or text context, while 30.80% have both.

Figure 3: Question length distribution (a) and context distribution in SCIENCEQA (b).

**Question analysis.** SCIENCEQA has a diverse set of science questions. Figure 2 shows a distribution of the first four words in the question text. A large number of question lengths and formats highlight the diversity of SCIENCEQA. The question lengths range from 3 words to 141 words, and the questions in SCIENCEQA have an average length of 12.11 words. The question length distribution is visualized against other VQA datasets in Figure 3 (a). As shown in the diagram, SCIENCEQA's distribution is flatter than other datasets, spanning more evenly across different question lengths.

**Context analysis.** Figure 3 (b) shows the number and percentage of questions with either an image context, a text context, or both. There are a total of 7,803 unique image contexts and 4,651 unique text

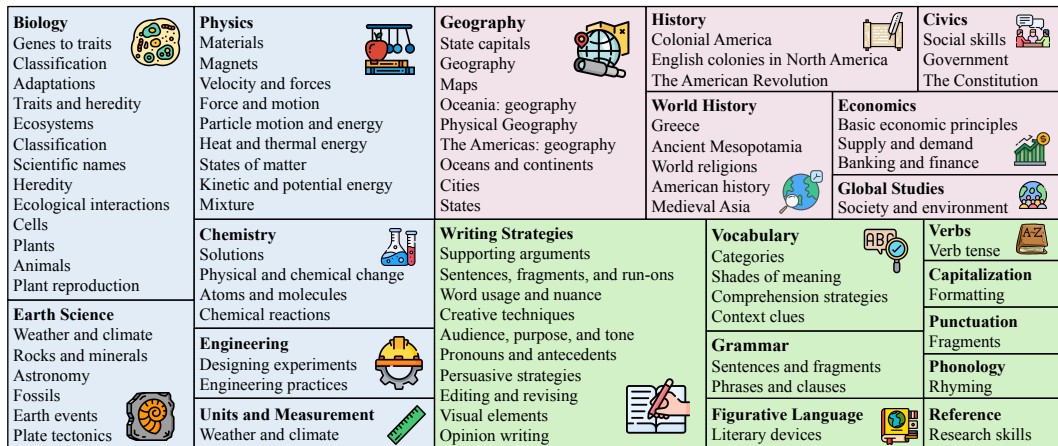

Figure 4: Domain diversity in SCIENCEQA. Each color corresponds to one subject: natural science, social science, and language science. For visual clarity, only the most frequent classes are shown.

contexts. 66.11% of the questions have at least one type of context information. The image context is in the format of diagrams or natural images, which visualize the critical scenario necessary for question answering or simply illustrate the question for better understanding. Similarly, the textual context can provide either semantically rich information or a simple hint to the question. Therefore, models need to be flexible and general to understand these diverse types of contexts.

**Domain diversity.** Each SCIENCEQA question belongs to one of the three subjects: natural science, social science, and language science. With each subject, questions are categorized first by the topic (*Biology*, *Physics*, *Chemistry*, etc.), then by the category (*Plants*, *Cells*, *Animals*, etc.), and finally by the specific skill (*Classify fruits and vegetables as plant parts*, *Identify countries of Africa*, etc.). SCIENCEQA has a total of 26 topics, 127 categories, and 379 skills. The treemap in Figure 4 visualizes the different subjects, topics, and categories and shows that SCIENCEQA questions are very diverse, spanning a wide range of domains.

## 3.2 Comparisons with Existing Datasets

Table 2 shows a comparison of SCIENCEQA and other science problem datasets. As shown in the table, SCIENCEQA is much larger than most other datasets. SCIENCEQA also has the largest set of images, spans across all 12 grades, contains the longest questions, and has the most diverse input sources. As opposed to limiting the subject to only natural science, SCIENCEQA also includes social science and language science, largely adding to the domain diversity of the dataset. Furthermore, most of the questions in SCIENCEQA are annotated with textual lectures (83.9%) and explanations (90.5%), which reveal the reasoning path to the correct answer. To the best of our knowledge, SCIENCEQA is the first large-scale multimodal science question dataset that annotates the answers with detailed lectures and explanations.

| | #Q | #I | AvgQ | MaxQ | Grades | Science subjects | Contexts | Images | Lecture | Explanation |
|---|---|---|---|---|---|---|---|---|---|---|
| Geometry3K [31] | 3,002 | 2,342 | 10.1 | 46 | 6-12 | natural (geometry) | image | diagram | ✘ | ✘ |
| AI2D [17] | 4,563 | 4,903 | 9.8 | 64 | 1-6 | natural | image | diagram | ✘ | ✘ |
| FOODWEBS [24] | ≈5,000 | ≈5,00 | - | - | 8 | natural (foodweb only) | image | diagram | ✘ | ✘ |
| ARC [5] | 7,787 | 0 | **20.4** | 128 | 3-9 | natural | ✘ | ✘ | ✘ | ✘ |
| TQA [18] | **26,260** | 3,455 | 9.2 | 57 | 6-8 | natural | image, text | diagram | ✔ | ✘ |
| IconQA [35] | 107,439 | 96,817 | 8.4 | 73 | PreK-3 | math | visual | diagram | ✘ | ✘ |
| WorldTree [13] | 1,680 | 0 | - | - | 3-5 | natural | ✘ | ✘ | ✘ | ✔ |
| OpenBookQA [37] | 5,957 | 0 | 10.6 | 68 | 1-6 | natural | ✘ | ✘ | ✘ | ✔ |
| QASC [20] | 9,980 | 0 | 8.0 | 25 | 1-9 | natural | ✘ | ✘ | ✘ | ✔ |
| **SCIENCEQA (ours)** | 21,208 | **10,332** | 12.1 | **141** | **1-12** | natural, social, language | image, text | natural, diagram | ✔ | ✔ |

Table 2: Statistics for SCIENCEQA and comparisons with existing datasets. #Q: number of questions, #I: number of images, AvgQ: average question length; MaxQ: maximum question length.

# 4 Baselines and Chain-of-Thought Models

In this section, we establish baselines and develop two chain-of-thought models on SCIENCEQA.

## 4.1 Baselines

**Heuristic baselines.** The first heuristic baseline is *random chance*: we randomly select one from the multiple options. Each trial is completed on the whole test set, and we take three different trials for an average result. The second heuristic baseline is *human performance*. We post the task to Amazon Mechanical Turk and ask workers to answer SCIENCEQA questions. Only workers who obtain a high school or higher degree and pass the qualification examples are qualified for the study. Each worker needs to answer a set of 10 test questions, and each question is answered by three different workers. For more details of the human performance study, see Appendix B.2.

**Zero-shot and few-shot baselines.** We establish the zero-shot baselines on top of UnifiedQA [19] and GPT-3 [4]. The zero-shot setup follows the format of QCM→A where the input is the concatenation of tokens of the question text (Q), the context text (C), and multiple options (M), while the output is to predict the answer (A) from the option set. We extract the caption from the captioning model based on ViT [7] and GPT-2 [47] for the image as the visual context. In the few-shot setting, we follow the standard prompting [4] where in-context examples from the training set are concatenated before the test instance. These in-context examples serve as an instruction for the language model to adjust to the specific task in SCIENCEQA.

**Fine-tuning baselines.** We first consider the fine-tuning baselines from VQA models [1, 21, 55, 9, 22, 35, 26] proposed in recent years. These VQA baselines take the question, the context, and choices as the textual input, take the image as the visual input, and predict the score distribution over choice candidates via a linear classifier. In addition, we build the fine-tuning baseline on top of the large language model UnifiedQA [19]. UnifiedQA takes the textual information as the input and outputs the answer option. Similarly, the image is converted into a caption that provides the visual semantics for the language model.

## 4.2 Language Models with the Chain of Thought

*A chain of thought* refers to a coherent flow of sentences that reveals the premises and conclusion of a reasoning problem [54]. A chain of thought clearly decomposes a multi-hop reasoning task into intermediate steps instead of solving the task in a black-box way. The chain of thought can be the step-by-step thought process [54] before arriving at the final answer or explanations [41] that come after the answer. The annotated lectures and explanations in SCIENCEQA serve as *demonstrations* of the chain of thought that mimics the multi-step reasoning steps of human beings. In this paper, we study if large language models can generate reasonable explanations as the chain of thought to reveal the thought process when answering SCIENCEQA questions. Further, we explore how the chain of thought can improve the reasoning ability of language models on SCIENCEQA in both few-shot and fine-tuning learning.

**UnifiedQA with the chain of thought.** UnifiedQA [19] is a state of the art model for multi-option question answering. The original architecture of UnifiedQA takes the question and options as the input and outputs a short phrase as the final answer. We make a format modification to develop UnifiedQA with the chain of thought (CoT), i.e., UnifiedQA is fine-tuned to generate a long sequence of text which consists of the answer followed by the lecture and explanation.

**GPT-3 via chain-of-thought prompting.** Recent research work [4, 38, 34] has shown that GPT-3 [4] can perform various tasks when provided with in-context examples in a standard prompt. Take multi-option question answering as an example, the standard prompt [36, 57, 29] builds instructions using in-context examples with components of the question text, options, and the correct answer text. This style of few-shot learning enables the GPT-3 model to answer specific questions without parameter updates. Different from standard prompting, we build GPT-3 via chain-of-thought (CoT) prompting, as shown in Figure 5. To be specific, for each test problem $t$, we map the prompt instruction $I : \{I_i\}_n, I_t$ into a textual format where $\{I_i\}_n$ refers to the instruction set of $n$-shot in-context examples from the training set, while $I_t$ denotes the test instruction. Instead of the way where the explanation comes before the answer [54], we feed the instruction $I$ into the encoder-

```
Question: question : $I_i^{ques}$
Options: (A) option : $I_{i1}^{opt}$ (B) option : $I_{i2}^{opt}$ (C) option : $I_{i3}^{opt}$
Context: context : $I_i^{cont}$
Answer: The answer is answer : $I_i^a$. BECAUSE: lecture : $I_i^{lect}$  explanation : $I_i^{exp}$

Question: question : $I_t^{ques}$
Options: (A) option : $I_{t1}^{opt}$ (B) option : $I_{t2}^{opt}$ (C) option : $I_{t3}^{opt}$ (D) option : $I_{t4}^{opt}$
Context: context : $I_t^{cont}$
Answer:
```

Figure 5: Prompt instruction encoding for the test example $t$ in GPT-3 (CoT). The prompt above consists of the instruction $\{I_i\}_1$ for the 1-shot training example and $I_t$ for the test example.

decoder model GPT-3 to generate the answer $a$ followed by the lecture $lect$ and explanation $exp$: $M : \{I_i\}_n, I_t \rightarrow a, lect, exp$.

## 5 Experiments

### 5.1 Experimental Setup

**Evaluation metrics.** The heuristics and VQA baselines treat our SCIENCEQA task as a multi-class classification problem with multiple options and are evaluated with the accuracy metrics. UnifiedQA and GPT-3 treat SCIENCEQA as a text generation problem. So the most similar option is selected as the final prediction to evaluate the question answering accuracy. The generated lectures and explanations are evaluated by automatic metrics [44, 28, 49] and human scores by annotators.

**Implementation details.** The VQA baselines are trained for a maximum number of 50 epochs with a learning rate of $5e-5$. We fine-tune the UnifiedQA for $50k$ iterations and evaluate every $1k$ iteration. The training process is stopped following the early stopping strategy with a patience period of three evaluations. For GPT-3, we use the `text-davinci-002` engine, which is the most capable model version suggested in the official documentation. More details can be found in Appendix B.1.

### 5.2 Results for Question Answering

Table 3 demonstrates the empirical results for Science Question Answering.

**VQA baselines.** We feed the VQA baseline models with the input of QCM format to predict answers A. Out of all the VQA models we benchmarked, VisualBERT [26, 27] performs the best on average (61.87%). Interestingly, Patch-TRM [35] beats VisualBERT in natural science (NAT) and language science (LAN), and it also performs better in higher-grade questions (67.50% *v.s.* 59.92%). However, in the subject of social science (SOC), VisualBERT outperforms Patch-TRM by a large margin (+22.39%). Such drastic changes in performance might imply that current VQA models are not generalized to process the challenging questions in SCIENCEQA.

**Language models.** We evaluate whether large-scale pretraining on text can help language models learn scientific knowledge and thus perform better on the SCIENCEQA task. For this purpose, we have tried two of the state-of-the-art pre-trained language models: UnifiedQA and GPT-3.

(i) **UnifiedQA.** The results show that without any supervised fine-tuning (zero-shot), UnifiedQA cannot beat any VQA baseline model, while the pretraining does help the model obtain some scientific knowledge to outperform the random baseline. When fine-tuned with the answer labels in SCIENCEQA, UnifiedQA$_{BASE}$ reports an accuracy of 70.12% on average. By further teaching the model to generate the answer along with lecture and explanation, the developed language model with chain-of-thought (UnifiedQA$_{BASE}$ (CoT)) brings additional improvements of +3.21% (QCM→AE) and +3.99% (QCM→ALE). These results show that generating the chain of thought along with the answer benefits the reasoning ability of language models.

(ii) **GPT-3.** The positive effect of pretraining is also proved by the surprisingly good results from GPT-3 in the same zero-shot setting as UnifiedQA. Without any fine-tuning, GPT-3 already reaches almost the best performance we can get. Interestingly, prompting the GPT-3 with two training examples with only answers results in a negligible difference. However, if we prompt GPT-3 with chain-of-thought prompting (QCM→ALE), we obtain the state-of-the-art result so far (75.17%).

| Model | Learning | Format | NAT | SOC | LAN | TXT | IMG | NO | G1-6 | G7-12 | Avg |
|---|---|---|---|---|---|---|---|---|---|---|---|
| Random chance | - | M→A | 40.28 | 46.13 | 29.25 | 47.45 | 40.08 | 33.66 | 39.35 | 40.67 | 39.83 |
| Q only [1] | train set | Q→A | 41.34 | 27.22 | 47.00 | 41.79 | 35.15 | 44.60 | 39.28 | 40.87 | 39.85 |
| $C_I$ only [1] | train set | $C_I$→A | 41.34 | 29.25 | 45.45 | 42.33 | 36.09 | 42.93 | 39.21 | 41.07 | 39.87 |
| Q+M only [1] | train set | QM→A | 52.66 | 51.86 | 60.18 | 55.57 | 50.37 | 57.42 | 52.53 | 57.88 | 54.44 |
| Q+$C_T$+M only [1] | train set | $QC_T$M→A | 57.28 | 49.04 | 61.36 | 60.46 | 52.80 | 58.82 | 54.44 | 60.51 | 56.61 |
| Q+$C_I$+M only [1] | train set | $QC_I$M→A | 58.97 | 53.77 | 60.45 | 62.85 | 54.49 | 57.63 | 56.72 | 61.04 | 58.26 |
| MCAN [55] | train set | QCM→A | 56.08 | 46.23 | 58.09 | 59.43 | 51.17 | 55.40 | 51.65 | 59.72 | 54.54 |
| Top-Down [1] | train set | QCM→A | 59.50 | 54.33 | 61.82 | 62.90 | 54.88 | 59.79 | 57.27 | 62.16 | 59.02 |
| BAN [21] | train set | QCM→A | 60.88 | 46.57 | 66.64 | 62.61 | 52.60 | 65.51 | 56.83 | 63.94 | 59.37 |
| DFAF [9] | train set | QCM→A | 64.03 | 48.82 | 63.55 | 65.88 | 54.49 | 64.11 | 57.12 | 67.17 | 60.72 |
| ViLT [22] | train set | QCM→A | 60.48 | 63.89 | 60.27 | 63.20 | 61.38 | 57.00 | 60.72 | 61.90 | 61.14 |
| Patch-TRM [35] | train set | QCM→A | 65.19 | 46.79 | 65.55 | 66.96 | 55.28 | 64.95 | 58.04 | 67.50 | 61.42 |
| VisualBERT [26, 27] | train set | QCM→A | 59.33 | 69.18 | 61.18 | 62.71 | 62.17 | 58.54 | 62.96 | 59.92 | 61.87 |
| UnifiedQA$_{SMALL}$ [48] | zero-shot | QCM→A | 47.78 | 40.49 | 46.00 | 50.24 | 44.12 | 44.39 | 45.56 | 46.21 | 45.79 |
| UnifiedQA$_{BASE}$ [48] | zero-shot | QCM→A | 50.13 | 44.54 | 48.18 | 53.08 | 48.09 | 46.69 | 47.58 | 50.03 | 48.46 |
| UnifiedQA$_{SMALL}$ [48] | train set | QCM→A | 53.77 | 58.04 | 61.09 | 52.10 | 51.51 | 61.46 | 58.22 | 53.59 | 56.57 |
| UnifiedQA$_{BASE}$ [48] | train set | QCM→A | 68.16 | 69.18 | 74.91 | 63.78 | 61.38 | 77.84 | 72.98 | 65.00 | 70.12 |
| **UnifiedQA$_{BASE}$ (CoT)** | train set | QCM→AE | 70.60 | 74.02 | 78.36 | 65.69 | 64.80 | 81.53 | 75.48 | 69.48 | $73.33_{3.21↑}$ |
| **UnifiedQA$_{BASE}$ (CoT)** | train set | QCM→ALE | 71.00 | 76.04 | 78.91 | 66.42 | 66.53 | 81.81 | 77.06 | 68.82 | $74.11_{3.99↑}$ |
| GPT-3 [4] | zero-shot | QCM→A | 75.04 | 66.59 | 78.00 | 74.24 | 65.74 | 79.58 | 76.36 | **69.87** | 74.04 |
| GPT-3 [4] | 2-shot | QCM→A | 74.64 | 69.74 | 76.00 | 74.44 | 67.28 | 77.42 | 76.80 | 68.89 | 73.97 |
| **GPT-3 (CoT)** | 2-shot | QCM→AE | **76.60** | 65.92 | 77.55 | **75.51** | 66.09 | 79.58 | **78.49** | 67.63 | $74.61_{0.64↑}$ |
| **GPT-3 (CoT)** | 2-shot | QCM→ALE | 75.44 | **70.87** | **78.09** | 74.68 | **67.43** | 79.93 | 78.23 | 69.68 | $75.17_{1.20↑}$ |
| Human | - | QCM→A | 90.23 | 84.97 | 87.48 | 89.60 | 87.50 | 88.10 | 91.59 | 82.42 | 88.40 |

Table 3: Evaluation of baselines over different classes in accuracy (%). Model names: Q = question, M = multiple options, C = context, $C_T$ = text context, $C_I$ = image context, CoT = chain of thought. Format names: A = answer, AE = answer with explanation, ALE = answer with lecture and explanation. Question classes: NAT = natural science, SOC = social science, LAN = language science, TXT = text context, IMG = image context, NO = no context, G1-6 = grades 1-6, G7-12 = grades 7-12. Segments 1: Random chance; Segment 2: Ablation studies on top of Top-Down; Segment 3: VQA baselines; Segment 4: UnifiedQA baselines and UnifiedQA with CoT; Segment 5: GPT-3 baselines and GPT-3 with CoT; Segment 6: Average human performance.

**Question:** What do these two changes have in common?\n
bleaching clothes\n
a piece of apple turning brown\n
**Options:** (A) Both are only physical changes. (B) Both are caused by cooling. (C) Both are chemical changes. (D) Both are caused by heating.\n
**Answer:** **The answer is C. BECAUSE:** A **physical change** is a change to a substance that does not change what the substance is. A **physical change** only changes the substance's physical properties, such as its size, shape, or color.\n
A **chemical change** is a change to a substance that changes what the substance is. A **chemical change** changes the substance's chemical properties, such as its chemical structure.\n
**The changes that you listed are both chemical changes.**\n
When you **bleach clothes**, the chemical structure of the fabric changes. This change makes the fabric less likely to hold color.\n
When **a piece of fruit turns brown**, the chemical structure of the fruit changes. This change makes the fruit taste different.

Figure 6: One example of the predicted answer along with the chain of thought from GPT-3 (CoT).

**Human performance.** Humans outperform all benchmarks consistently across question classes, context types, and grades, *e.g.,* a 20.07% gap for questions with the image context (IMG) between humans and our best performing model. The gap is to be filled by future research on multimodal reasoning for scientific question answering.

## 5.3 Results for Generated Explanations

One prediction example of GPT-3 (CoT) is visualized in Figure 6. We can see that GPT-3 (CoT) predicts the correct answer and generates a reasonable lecture and explanation to mimic the human thought process. We further report automatic metrics (BLEU-1/4 [44], ROUGE-L [44], and (sentence)

Similarity [49] to evaluate the generated lectures and explanations, as shown in Table 4. The Similarity metric computes the cosine-similarity of semantic embeddings between two sentences based on the Sentence-BERT network [49]. The results show that UnifiedQA$_{BASE}$ (CoT) generates the most similar explanations to the given ones. However, it's commonly agreed that automatic evaluation of generated texts only provides a partial view and has to be complemented by a human study. By asking annotators to rate the relevance, correctness, and completeness of generated explanations, we find that the explanations generated by GPT-3 (CoT) conform best to human judgment.

| Model | Format | BLEU-1 | BLEU-4 | ROUGE-L | Similarity | Relevant | Correct | Complete | Gold |
|---|---|---|---|---|---|---|---|---|---|
| UnifiedQA$_{BASE}$ (CoT) | QCM→ALE | **0.397** | **0.370** | **0.714** | **0.811** | 80.4% | 76.6% | 76.1% | 56.9% |
| GPT-3 (CoT) | QCM→AE | 0.234 | 0.048 | 0.351 | 0.561 | 76.9% | 73.0% | 70.5% | 52.5% |
| GPT-3 (CoT) | QCM→ALE | 0.192 | 0.052 | 0.323 | 0.595 | **88.5%** | **78.8%** | **84.5%** | **65.2%** |

Table 4: Automatic metrics (BLEU-1/4, ROUGE-L, Similarity) and human evaluation of generated explanations. Note that a gold explanation refers to one that is relevant, correct, and complete.

## 5.4 Analysis

**Blind studies.** Blind studies are conducted on top of the modification of the full model, Top-Down [1]. The results achieved in blind studies of Q only and C$_I$ only are close to random chance, showing that the SCIENCEQA dataset is robust and reliable in distribution. The performance drops in Q+M only, Q+C$_T$+M only, and Q+C$_I$+M only indicate that all input components provide critical information for answering SCIENCEQA questions.

**Prompt types.** We study the effect of prompt types and visualize the comparison in Figure 7 (a). It shows that prompting the GPT-3 model with both lectures and explanations (QCM→ALE) results in the highest accuracy on average and the smallest variance. In contrast, prompting with only explanations (QCM→AE) gives the largest variance, resulting in a less stable model.

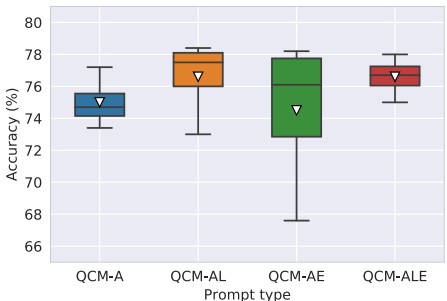

(a) Acc. v.s. different prompts with 4-shot examples.

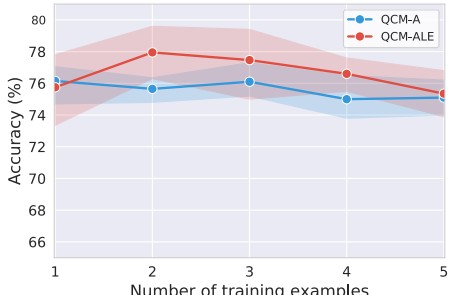

(b) Acc. v.s. different # of training examples.

Figure 7: Accuracy of GPT-3 (CoT) cross different prompt types (a) and # of training examples (b).

**Number of in-context examples.** In Figure 7 (b), we further investigate how different numbers of training examples encoded in prompts can affect the prediction accuracy. The QCM→ALE prompt type outperforms or performs comparably to the QCM→A type with all numbers of examples. And we observe the peak performance of QCM→ALE with 2 training examples being prompted. After that, the accuracy goes down as more training examples are added to the model.

**Dynamic sampling.** In Table 5, instead of random sampling, we try to dynamically select the in-context examples to prompt with the same class as the test sample. However, slight differences in prediction accuracy are observed when comparing them to simple random sampling.

| Prompt type | Sampling | Acc. (%) |
|---|---|---|
| QCM→ALE | Dynamic (same topic) | 75.15 |
| QCM→ALE | Dynamic (same category) | 74.58 |
| QCM→ALE | Dynamic (same skill) | 75.10 |

Table 5: Dynamic sampling for GPT-3 (CoT).

**Upper bound.** We search the upper bound of the GPT-3 accuracy by feeding the gold lecture and explanation in the test prompt. As reported in Table 6, QCME*→A outperforms the QCM→ALE baseline by 18.86% and QCMLE*→A outperforms QCM→ALE by 18.96%, indicating a potential improvement direction by generating correct explanations before answering science questions.

| Prompt type | Sampling | Acc. (%) |
|---|---|---|
| QCML*→A | Random | 73.59 |
| QCML*→AE | Random | 74.32 |
| QCME*→A | Random | $94.03_{18.86\uparrow}$ |
| QCMLE*→A | Random | $\mathbf{94.13}_{18.96\uparrow}$ |
| QCM→ALE | Random | 75.17 |

Table 6: Upper bound of GPT-3 (CoT).

| Prompt type | Sampling | Acc. (%) |
|---|---|---|
| QCM→LA | Random | 60.6 |
| QCM→EA | Random | 56.0 |
| QCM→LEA | Random | 55.4 |
| QCM→ELA | Random | 51.5 |
| QCM→ALE | Random | **73.6** |

Table 7: Different positions of L/E for GPT-3 (CoT).

**Positions of lectures and explanations.** We study the performance of GPT-3 (CoT) in terms of different positions of lectures and explanations on 1,000 test examples. The results are shown in Table 7. There could be huge accuracy decreases if GPT-3 (CoT) predicts lectures and explanations before answers. It is mainly because if GPT-3 (CoT) is formulated to generate the long lecture and explanation first, there is a greater chance that it will stop generating the prediction early or use up the maximum token limits before obtaining the required answer.

**CoT learns with fewer data.** To study if the chain of thought helps language models learn more efficiently, we report the accuracies of UnifiedQA and UnifiedQA (CoT) fine-tuned on different sizes of the training set in Figure 8. UnifiedQA (CoT) benefits language models by learning the coherent reasoning path when answering questions, resulting in similar accuracy with fewer training examples.

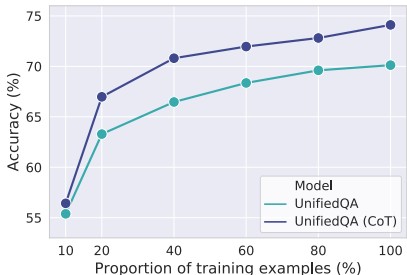

Figure 8: UnifiedQA (CoT) learns efficiently with fewer training examples.

**Error analysis.** GPT-3 via chain-of-thought prompting obtains promising results but still fails to answer a wide range of challenging questions in SCIENCEQA. See examples of failure cases in Appendix B.4. The failure cases can be classified into two types: (a) the model fails to understand the multimodal inputs and lacks domain-specific knowledge to arrive at the correct answer; (b) the model generates the wrong chain of thought with irrelevant, incorrect, or incomplete information.

## 6  Discussion and Conclusion

In this paper, we propose SCIENCEQA, a dataset that features 21,208 multi-option questions with multimodal contexts from the science curriculum. To the best of our knowledge, SCIENCEQA is the first large-scale multimodal science dataset where most questions are annotated with corresponding lectures and explanations. We establish various baselines, including recent VQA models and large language models on SCIENCEQA. We further study if language models can generate reasonable explanations and then benefit the reasoning ability. Experiments show that UnifiedQA with the chain of thought can achieve an improvement of 3.99% and few-shot GPT-3 via chain-of-thought (CoT) prompting can obtain a satisfactory accuracy of 75.17% on SCIENCEQA. 65.2% of the generated explanations from GPT-3 (CoT) meet the gold standard by human evaluations.

## 7  Acknowledgment

We would like to thank the anonymous reviewers for their valuable comments and suggestions. We would also like to thank Xiaodan Liang for insightful discussions on dataset collection. We thank our colleagues at The Allen Institute of AI (AI2), Jiasen Lu and Jungo Kasai for helpful discussions. The work does not relate to Liang Qiu's position at Amazon Alexa.

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
