# OpenReview forum: "Learn to Explain: Multimodal Reasoning via Thought Chains for Science Question Answering"
_NeurIPS.cc/2022/Conference — NeurIPS 2022 Accept_

### Official Review · Reviewer_E8dv · 2022-07-04

**Rating:** 7
**Confidence:** 4
**Soundness:** 3 good
**Presentation:** 4 excellent
**Contribution:** 3 good

**Summary:**

This paper aims to aid the development of a reliable model that is capable of generating a coherent chain of thought (CoT) when arriving at the correct answer to reveal the multi-step reasoning process.  Since existing science question datasets either lack annotated explanations for the answers or are restricted to the text modality with small data scales and limited topics, the paper presents Science Question Answering (SQA), a large-scale multi-choice dataset that contains multimodal questions with explanations and features rich domain diversity. It further designs language models to learn to generate lectures and explanations as to the chain of thought to mimic the reasoning process. Experimental results show that CoT benefits large language models in both few-shot and finetuning learning. And analysis shows that CoT helps language models learn from fewer data.


**Questions:**

1. Why aren't all examples annotated with lectures and explanations?
2. Why are CoTs lectures and explanations instead of explanations? Shouldn't the lecture be part of the model's input? Are there any experimental results for QCML->A and QCML->AE?
3. In table 3, why does GPT perform worse in the 2-shot setting than in the zero-shot setting?


**Limitations:**

The authors have discussed the limitations of this paper.

**Strengths And Weaknesses:**

Strengths:
1. The presentation is good. The paper is easy to read.
2. The proposed datasets may aid the development of model reasoning in science QA.
3. Experiments are extensive and enlightening.

Weaknesses:
1. The novelty of the proposed dataset and baselines are limited.
2. The evaluation of the generated CoTs (lectures and explanations) still relies heavily on human judgment. Since the automatic metrics used in the paper are less consistent with human judgment, better automatic evaluation methods specifically for CoT are desired.
3. The correlation between question answering performance and explanation generation performance is not discussed.

---

> ### Author Response · Authors · 2022-08-02
> **Responses to Reviewer E8dv (R5)**
>
> Dear reviewer, thank you for the insightful comments. We appreciate your time and effort. We are glad that you recognize our large-scale dataset features rich domain diversity and aids the development of model reasoning in related areas. We address your questions below.
>
> > **W1: The novelty of the proposed dataset and baselines.**
>
> Please see the first point, “**Novelties and contributions of our work**” in the **general responses to all reviewers** above.
>
> > **W2: Better automatic evaluation methods specifically for CoT.**
>
> It is really a good point to introduce new automatic evaluation methods specifically for CoT that are able to obtain consistent evaluation results as human evaluators. Existing automatic evaluation metrics such as BLEU-1, ROUGE-L, and Sentence Similarity prefer "similar" generations to the training data, while human beings prefer predictions that are relevant, correct, and complete. Thanks for your valuable suggestions! We are looking forward to more consistent automatic evaluation methods in the follow-up research work.
>
> > **W3: Discussion of the correlation between question answering performance and explanation generation performance.**
>
> Overall, the correlation between question answering performance and explanation generation performance is positive. For example, 2-shot GPT-3 (CoT) with the QCM→ALE prompting type achieves the highest QA accuracy and, at the same time, obtains the highest proportion of gold explanations among all the predictions. But we do have some special cases where the generated explanation is gold while the predicted answer is wrong, and vice versa. These failure cares are well discussed in Appendix B.4 for case study and limitations.
>
> > **Q1: Why aren't all examples annotated with lectures and explanations?**
>
> We automatically collected the raw data for the SQA dataset from online resources. Lectures and explanations could be missing due to the lack of annotations. We kept the examples without lectures or explanations to maximize the domain diversity of the constructed SQA dataset.
>
> > **Q2a: Why are CoTs lectures and explanations instead of explanations? Shouldn't the lecture be part of the model's input?**
>
> We believe that lectures, providing external background knowledge, are a general kind of explanation for the questions and answers. We conducted extensive ablation studies for different prompting formats and methods and further discussed the results in Section 5.4 (Table 6). It is found that the format of QCM→ALE performs the best among various settings.
>
> We studied the upper bound of GPT-3 (CoT) by feeding the lecture or solution to the model's input. The results show that with the gold lecture in the input, QCML*→AE performs worse than QCM→ALE with an accuracy decrease of 1.58%. It is probably because, in pre-training data such as science and math textbooks, solutions appear in the output after the answer, so this representation (QCM→ALE) makes it familiar to the model.
>
> > **Q2b: Are there any experimental results for QCML->A and QCML->AE?**
>
> Yes, we had these results. We added them to the revised paper in Table 7.
>
> > **Q3: In Table 3, why does GPT perform worse in the 2-shot setting than in the zero-shot setting?**
>
> For the standard GPT-3 models, the accuracy difference between these two settings is 0.07%, which could result from a random deviation. It shows that without the CoT prompting, GPT-3 might not gain benefits from the in-context examples in the SQA dataset. Please also note that a similar observation has been made in other works where it has been shown that adding examples does not really help in the in-context setting for GPT-3 [23].
>
> [23] Prompt Programming for Large Language Models: Beyond the Few-Shot Paradigm, 2021

---

### Official Review · Reviewer_ACnG · 2022-07-09

**Rating:** 6
**Confidence:** 3
**Soundness:** 3 good
**Presentation:** 3 good
**Contribution:** 3 good

**Summary:**

In this work, the authors propose a new multi-modal multiple-choice question answering dataset for science questions, with explanations. The dataset covers multiple domains, and has questions with text and image contexts.

The authors evaluate several baselines in different setups: VQA, fine-tuning UnifiedQA, prompting GPT-3 and find that the explanations provided in the dataset lead to improved accuracy on the dataset, both when using in a fine-tuning context as well as when used in prompting GPT-3.


**Questions:**

Q1: what is the evidence that images are actually necessary for answering the questions in VQA experiments? Do you have questions referring to an image without having textual context that contains all the necessary information? Can you include at least one such example in the paper somewhere because I’m not sure the baby example is clear in this regard?

Q1.a: are questions that require an image to be answered included in the evaluation of text-only models and why does that make sense? How do we know the captions capture the necessary information?

Q2: why no QCM → (L)EA?

Q3: questions (i) - (iii) regarding table 3 (see above)

Q4: what is the variance on the reported numbers?

Q5: Are you always using a caption model ? How is the caption added to the context? How is the caption model trained? What is the effect of including generated captions for your text-based experiments? I think the captioning part should be explained in more detail, and evaluated in an ablation. Are these captions provided in the dataset for use by others?

Small notes:
* table 1 columns should be explained
* l. 82: “multi-model”?
* l. 97: “for examples”?
* l. 97-99: sentence incomplete
* l. 219: “will be stopped” → “is stopped”
* l. 246: “on SQA so far (75.17%) on SQA”
* Fig 3,a: the lines are a bit too thick, it is a bit difficult to read this figure
* Fig 3,b: I think this figure is unnecessary because it repeats what was already mentioned in the text, and in table 2, and takes up a lot of space relative to the contained info (unlike Fig. 3,a for example). I would add the % of questions without any context in Table 2 though.


**Limitations:**

QCM→(L)EA not evaluated and not clear why.

CoT not evaluated with VQA.

The authors includes an impact statement in appendix.


**Strengths And Weaknesses:**

## Strengths:
* (mostly) well-written
* new dataset appears to fill a few gaps from existing datasets
* extensive evaluation with VQA models, as well as text-based finetuned and prompt-based models (UnifiedQA and GPT-3)

## Weaknesses:
* mediocre originality, not clear what really new questions or lessons
* confusing evaluation presentation and confusing to use
* some experimental settings are not explored (QCM→(L)EA)
* a round of proof-reading is needed to fix some mistakes and improve sentence construction
* I think it needs renaming because SQA already exists: https://paperswithcode.com/dataset/sqa

## Detailed review:
The paper is mostly well-written, and presents an interesting dataset for the QA community, and establishes reasonable baselines in various scenarios. Even though the contribution is technically solid (once a few concerns are addressed), there doesn’t appear much new to be learned from this work. It appears to be an interesting dataset for science QA, in particular because it provides explanations. The problem is rather that it’s easy to dismiss this work as showing that chain-of-thought based training works better with some baselines in a similar set-up to what was already explored for different domains. The addition of the visual modality is unfortunately not tied to CoT experiments and appears as another loose dimension of the dataset.
Some concerns that I had while reading the work were that the way evaluation results are presented is quite confusing, which could lead to confusion in practical use by others.
All main results (VQA and text-based) are crammed into Table 3 and there are several issues: (i) the segments in the table are not labeled (ii) when evaluating VQA models, are all the questions used (even the ones without image context, and does this make sense)? (iii) what do the TXT and IMG columns mean? What about questions without either?
I think the main evaluation needs more explanation and improved presentation.
Then, some important settings are missing from evaluation, in particular QCM → (L)EA (generating explanation before answer) hasn’t been included. I think there could be a significant difference between QCM→A(L)E and QCM→(L)EA for auto-regressive decoders because the reasoning steps generated before producing the answer can contribute to predicting the answer more correctly (see also Fig. 5 in [48]).
In addition, it would have been nice to provide some numbers for VQA with CoT. The currently tested VQA models are using a classifier to choose between the multiple choices, but it seems a baseline can be made by replacing that with a decoder? I think improvements in VQA due to CoT could also be an interesting additional selling point for this work.

Conclusion: if the above concerns regarding evaluation presentation and missing evaluation settings are addressed, I think it’s a solid work that presents an interesting dataset for the QA community. However, I wish there were more interesting new questions raised or new lessons learned.

(Note: I tried to formulate all of these concerns in the questions section as well for easier answering)

---

> ### Author Response · Authors · 2022-08-02
> **Responses to Reviewer ACnG (R4) (Cont.)**
>
> > **Q3(i): The segments in Table 3 are not labeled.**
>
> The segments in Table 3 are listed below:
> - Segment 1: Random chance
> - Segment 2: Ablation studies on top of the Top-Down model
> - Segment 3: VQA baselines
> - Segment 4: UnifiedQA baselines and UnifiedQA with CoT
> - Segment 5: GPT-3 baselines and GPT-3 with CoT
> - Segment 6: Average human performance
>
> Thank you for your suggestion. We have added them to the revised version.
>
> > **Q3(ii): When evaluating VQA models, are all the questions used (even the ones without image context, and does this make sense)?**
>
> Yes. For the results in Table 3, we evaluate the questions of the whole test set, and the last column, "Avg" reports the average accuracy of all questions. To further analyze how different baselines perform across different classes, we also report the accuracy in the columns from 4 to 11. When evaluating VQA models, all the questions are used.
>
> Note that the VQA model can also operate without the image inputs. The embeddings for the image context input for those questions without image context are set to be zero vectors, which is widely used in previous VQA work when conducting blind studies [21,22].
>
> [21] VQA: Visual Question Answering, 2015
>
> [22] CLEVR: A Diagnostic Dataset for Compositional Language and Elementary Visual Reasoning, 2016
>
> > **Q3(iii): What do the TXT and IMG columns mean? What about questions without either?**
>
> The TXT column refers to results for a set of questions with a text context, while IMG refers to results for questions with an image context. We added the results for questions without either in the "No" column in the revised paper. For most baselines in Table 3, the performance gaps between most baselines and the random guess baseline are larger for questions without context than for those without context. It indicates that the text and image contexts in SQA are critical for answering the questions.
>
> > **Q4: What is the variance in the reported numbers?**
>
> We visualized the error bars for the ablation studies in Figure 6. Instead, as GPT-3 is expensive to run for repeated experiments on the entire test set, we only ran the reported experiments in ablation studies on a subset of the test set and ran the model with optimal parameters for the final model once, as reported in the main table (Table 3).
>
> > **Q5: Details of using the caption model.**
>
> We only used the caption model for text-only baselines and models. We adopted a caption model to generate a natural language description for the image and then concatenate the description with other texts as the inputs of the model. The image description can provide the semantic information in the image and further enables the text-only models to work on the SQA dataset. The caption model we used is a SOTA pre-trained image captioning model, and we applied an online API developed by Huggingface (https://huggingface.co/nlpconnect/vit-gpt2-image-captioning) to generate captions. These captions can be easily reproduced by others using this easy-to-use online tool. And we will also release the captions in the dataset and the generating scripts for easy reproduction.

---

> ### Author Response · Authors · 2022-08-02
> **Responses to Reviewer ACnG (R4) (Cont.)**
>
> > **Q1: What is the evidence that images are actually necessary for answering the questions in VQA experiments?**
>
> First, we conducted ablation studies to validate the importance of each input element on top of the Top-Down model and report the results in Table 3. Q+$C_T$ +M only refers to the Top-Down model that does not take the image context into the inputs. Results show that Q+$C_T$ +M achieves an accuracy of 52.80% for the questions with image context, a decrease of 2.06% compared to the full model (Top-Down).
>
> Second, visualization of the SQA examples (please check the Appendix or the visualization tool we provide in the supplementary for more information) shows that image contexts like charts, maps, and tables are critical and necessary to answer the questions correctly (e.g., “What is the capital of the country highlighted on this map?” cannot be answered without the image by definition).
>
> Third, all VQA baselines in Table 3 can achieve larger performance improvements from the random guess baseline for questions without context than for questions with image context. For example, VisualBERT achieves an improvement of 24.88% from 33.66% to 58.54% for questions without context (“NO”), while this improvement is only 20.09% for questions with image context (“​​IMG”). It shows that the image context poses additional challenges for existing VQA baselines. For example, other than natural images, there are diagrams with diverse formats in an image context, including tables, maps, and illustrations. Current VQA baselines are still struggling to understand the diagrams well because they are mostly trained on natural images.
>
>
> > **Q1.a: Are questions that require an image to be answered included in the evaluation of text-only models and why does that make sense? How do we know the captions capture the necessary information?**
>
> Yes. For fair comparisons, we need to make sure that different baselines are evaluated on the same set of questions. For example, the text-only models (e.g., Q+M only, UnifiedQA, GPT-3) are all evaluated on the whole test set, and the results for different classes are listed in the corresponding columns, respectively.
>
> The image context could provide either complementary information that is helpful for question answering or critical visual information that is necessary for the models. According to rough estimates, about 30% of questions feature critical images for question answering. Even though our GPT3 (CoT) is able to achieve promising average prediction performance, it fails many times because of some information gaps associated with the captioning systems. We apply the SOTA captioning model to extract the textual descriptions for GPT-3 (CoT). But this captioning model fails to capture fine-grained semantic information in images such as maps, diagrams, tables, and illustrations. Therefore, this issue with the captioning model results in the prediction failures for GPT-3 (CoT), which are discussed in Appendix B.4.
>
> One prospective solution to this problem is to propose unified vision-and-language large models to have a more powerful ability to capture both textual and visual information for question answering and explanation generation. We are glad to see that recent work [20] has been exploring this direction and has achieved promising results.
>
> [20] Unified-IO: A Unified Model for Vision, Language, and Multi-Modal Tasks, 17 Jun 2022.
>
>
> > **Q2: Why no QCM→(L)EA?**
>
> We had the result for QCM→(L)EA but unfortunately, we didn’t keep it in the submitted paper due to limited space. The result has been added to the revised paper now (please refer to Table 7)!

---

> ### Author Response · Authors · 2022-08-02
> **Responses to Reviewer ACnG (R4)**
>
> Dear reviewer, thank you for the insightful comments and we appreciate your time and effort.  We are excited that you evaluate that our dataset is interesting, covers multiple domains, and fills gaps in existing datasets. We are glad that you recognize we establish extensive baselines and our models lead to performance improvements in both fine-tuning and few-shot settings. We are willing to address your concerns below.
>
> If you have any further questions, please feel free to let us know. We are grateful to have the chance to resolve these questions with you in the discussion phase!
>
> > **W1: The originality of the work.**
>
> Please see the first point, “**Novelties and contributions of our work**” in the **general responses to all reviewers** above.
>
> > **W2: Details of evaluation.**
>
> The evaluation metrics and key implementation details are introduced in Section 5.1. More details can be found in Appendix B.1, where we discuss the fine-tuning settings, batch sizes, the newline character, the captioning model tool, compute resources, and GPT-3 settings. We will release the dataset, scripts, and prediction results for easy reproduction of our results when the camera-ready version is ready.
>
> > **W3: To explore more experimental settings.**
>
> Thank you for your suggestion! We did try more experimental settings, however, owing to the space limitation, we had to omit some exploration experiments that did not work. We have included more settings, including QCM→EA, QCM→LA, QCM→ELA, and QCM→LEA in the revised paper now (please refer to Table 6 and Table 7). The results are listed below as well:
> - QCM→EA, 56.0% (added)
> - QCM→LA, 60.6% (added)
> - QCM→ELA, 51.5% (added)
> - QCM→LEA, 55.4% (added)
> - QCM→ALE, 55.4% (final)
>
> > **W4: A round of proofreading of the paper.**
>
> We have incorporated your mentioned notes and fixed some typos we found in the revised paper. We will definitely do several rounds of careful proofreading to make sure the paper is technically sound before it is released.
>
> > **W5: Renaming the SQA dataset.**
>
> We really appreciate your helpful suggestions. We agree with you completely. We are considering renaming SQA as "ScienceQA"; fortunately, "ScienceQA" has not been used by released datasets on https://paperswithcode.com/datasets. To avoid confusion for other reviewers, we will tentatively keep the name SQA in the rebuttal phase and update it in the camera-ready version. Thanks again for your great efforts in reviewing our paper!

---

> > ### Comment · Reviewer_ACnG · 2022-08-08
> > **Response to authors**
> >
> > Thank you for the extensive and thoughtful reply and additional results. I think you addressed most of the issues I had with this work and would be happy to see it in the conference.

---

> > > ### Author Response · Authors · 2022-08-08
> > > **Thanks for your helpful comments!**
> > >
> > > Thank you so much for your great efforts and time!
> > >
> > > It is a great encouragement to our work, and we are glad to see that we have addressed most of your concerns. And many thanks for your feedback to make our paper stronger and more solid!

---

### Official Review · Reviewer_M8Qt · 2022-07-19

**Rating:** 6
**Confidence:** 4
**Soundness:** 2 fair
**Presentation:** 2 fair
**Contribution:** 2 fair

**Summary:**

The paper introduces a new dataset ScienceQA which consists of multiple choice questions with knowledge source and explanations. The dataset covers three domains of elementary through high school questions. The authors also conduct experiments using chain-of-thought methods, which prompts GPT3 and unified QA with a concatenation of question context, answer and lecture+explanation.

**Questions:**

Possible improvements directions:
1. I think the lectures could be retrieved rather than generated, assume that you have a large documents of lecture contents. Using a retrieval based model to retrieve the lectures and use the GPT-3 models to condition on the lecture and generate explanations, and finally the answers could boost the model performance.

2. Captioning system that converts images to text might be suboptimal, and could also explain why the accuracy for image questions seems to be significantly lower than the text questions. I think you might want to train end-to-end, or find a system that's focused on image description, rather than image captioning.

3. Report standard deviations in the main table. I think there are high variances for in-context learning, but the table doesn't show it.

**Ethics Review Area:**

["I don’t know"]

**Limitations:**

The authors discussed limitations but in a quite narrow and empirical way, would be nice to see broader, higher-level limitations of this class of methods.

**Strengths And Weaknesses:**

Strength:
1. The dataset looks promising, given that it covers both image and text modality and various science domains.

Weakness:
1. I don't understand why the proposed chain-of-thought (QCM-ALE) would be helpful when you generate answer before generating explanations (especially for the few-shot GPT3 examples). I think this might improve the quality of explanations, but given GPT3 is left-to-right autoregressive, the generated answers cannot take advantage of the generated explanations in the right contexts, which seemingly weakens the power of CoT, that looks like  QCM-LEA or QCM-ELA.

2. Should compare to QCM-LEA or QCM-ELA. I think generating answers conditioned on the explanations would probably yield more performance gains, because the answers can condition on the generated explanations.

3. The method is not novel enough. Other than my concerns (1) that the chain-of-thought is not used in its most powerful form: the chain-of-thought idea is not novel and nor is explanation/rationale generation.

---

> ### Author Response · Authors · 2022-08-02
> **Responses to Reviewer M8Qt (R3) (Cont.)**
>
> > **Q1: The lectures could be retrieved rather than generated.**
>
> Yes, it is a good point! The improvement direction that the lectures could be retrieved rather than generated is based on the assumption that lectures could be correctly retrieved and the retrieved lectures could further benefit the performance accuracy.
>
> However, our initial experiment results in Tables 5 and 6 indicate that retrieving lectures might not be helpful in the SQA dataset:
>
> - In Table 6, we study the upper bound of GPT-3 (CoT). We use gold and perfect lectures in the input (QCML*→A, QCML*→AE) and it is not helping. For example, the performance of QCML*→A decreases from 73.97% (QCM→A) to 73.59%. Retrieval will result in imperfect lectures, so we are not sure how it can help with SQA.
>
> - In Table 5, we investigate if GPT-3 (CoT) could benefit from more similar in-context examples. For example, if we adopt dynamic sampling with the same skill, it is more likely that the lectures in the in-context examples are similar to or the same as the lectures regarding the test example. But results show that similar lectures in the in-context examples would not improve the QA performance.
>
> > **Q2: Captioning system that converts images to text might be suboptimal. I think you might want to train end-to-end, or find a system that's focused on image description, rather than image captioning.**
>
> We only convert images to captions when evaluating large language models such as UnifiedQA and GPT-3. For VQA baselines, the raw images are fed into the neural networks. And it is found that current SOTA VQA baselines consistently perform worse when answering questions with visual context than questions with textual context.
>
> We respectfully disagree with you on “finding a system that's focused on image description, rather than image captioning.” In the area of vision-and-language learning, generating the description for an image is the same task as image captioning. For example, [16] defines the task of image captioning as “automatically generating a natural language description of an image”.
>
> We totally agree that large unified vision-language models for general purposes are good options for a better joint understanding of both image and text contexts. We believe that it is a worthwhile direction for follow-up research work [17].
>
> [16] Image Captioning With Semantic Attention, 2016
>
> [17] Unified-IO: A Unified Model for Vision, Language, and Multi-Modal Tasks, 17 Jun 2022.
>
>
> > **Q3: Report standard deviations in the main table.**
>
> The GPT-3 API is very expensive, and it is not practical to repeat the experiments multiple times on the whole test set over different settings in the main table. Instead, following recent work [18,19] on large models via prompting, we conduct extensive ablation studies on a subset of the test and report the final results for the models with optimal hyperparameters. For example, in Figure 6, we discuss the accuracy of GPT-3 (CoT) across different prompt types and the number of training examples with reported deviation ranges.
>
> [18] Self-Consistency Improves Chain of Thought Reasoning in Language Models, 2022
>
> [19] Large Language Models are Zero-Shot Reasoners, 2022
>
> > **Q4: Discussion of broader, higher-level limitations of this class of methods.**
>
> We appreciate your feedback on the broader discussion of this class of methods. Although large pre-trained language models such as GPT-3 can achieve amazing results on a lot of downstream tasks like science question answering, they still suffer from several limitations that researchers are working to address. First, they are often inconsistent—failure cases in the appendix show that you can get correct answers with wrong explanations. Second, GPT-3 is parroting the patterns in the huge amount of training data, which could lead to generation bias.

---

> > ### Comment · Reviewer_M8Qt · 2022-08-05
> > **Thanks for the additional experiments.**
> >
> > Thanks for the additional experiments and the nice discussions!
> >
> > The empirical results are quite convincing for putting answer before explanations! It would be nice to see that the models assign higher probabilities to answer-explanation than explanation answer to justify your point that this is more in-distribution.
> >
> > I have improved my rating to 6.

---

> > > ### Author Response · Authors · 2022-08-05
> > > **Thanks for your encouraging comments!**
> > >
> > > Thank you so much for improving your rating!
> > >
> > > Yes, language large models like GPT-3 could perform better in the few-shot setting if the provided in-context examples follow the in-distribution format of the training corpora than the out-distribution format. This phenomenon is also supported by recent work where the authors find the *human-like content effects*—language model predictions often reflect human knowledge and beliefs about the world.
> > >
> > > [24] Language models show human-like content effects on reasoning, https://arxiv.org/abs/2207.07051
> > >
> > > It is a great encouragement to our efforts, and we are happy to see that we have addressed your comments. We appreciate all your suggestions!

---

> ### Author Response · Authors · 2022-08-02
> **Responses to Reviewer M8Qt (R3) (Cont.)**
>
> > **W3a: The chain-of-thought is not used in its most powerful form.**
>
> We respectfully disagree with you on this. We conducted extensive ablation studies in Section 5.4 (Line 262 - 306) to discuss the performance regarding different numbers of in-context numbers, different prompting formats, and different sampling methods for GPT-3 (CoT) on SQA. Please take a look at Table 7 where we do other forms of CoT like QCM→LEA, QCM→ELA, etc. and they do not work. The analysis shows that  GPT-3 (CoT) performs best in the format of QCM→ALE with two in-context examples via dynamic sampling (same topic). Finally, we report the result of GPT-3 (CoT) with the optimal setting in Table 3.
>
> > **W3b: Novelty of the chain-of-thought idea.**
>
> The chain-of-thought reasoning in language models is still understudied after Wei et al. and Wang et al. first found that the chain of thought has the potential to improve the performance of complex QA tasks like math word problem-solving in 2022. In this paper, we extensively explore CoT prompting on SQA and show that CoT benefits large language models in both few-shot and fine-tuning learning by improving model performance and reliability via generating explanations. To be more specific,
>
> (1) CoT has been shown to be helpful in a few-shot setting (Wei et al.). We extend CoT to the fine-tuning setting and show that it improves the performance consistently.
>
> (2) CoT in the original paper [15] was shown to improve performance in the task of question answering. We show that, along with this performance, it also helps generate reliable explanations for the answer.
>
> (3) The original CoT paper [15] is in the text-only domain. We show that CoT and its variants also work in the multi-modal setting, provided we extract the visual information in natural language from the input images.
>
> (4) We also see evidence that CoT in our final setup (QCM→ALE) also helps the model learn from fewer data. This is analogous to how humans being taught with explanations learn quickly just with a few examples.
>
> (5) We further explore the upper bound of GPT-3 and find that if the GPT-3 model is fed with the gold lecture and explanation in the inputs.
>
> (6) We also investigate the effect of various parameters: (a) prompt format, (b) number of examples, and (c) sampling methods on model performance with CoT to better understand the sensitivity and statistical significance of the performance gains.
>
> [15] Chain of Thought Prompting Elicits Reasoning in Large Language Models, 2022

---

> ### Author Response · Authors · 2022-08-02
> **Responses to Reviewer M8Qt (R3)**
>
> Dear reviewer, we appreciate your time and thank you for your helpful comments. We address your concerns below.
>
> > **W1: Why the proposed chain-of-thought (QCM-ALE) would be helpful when generating answers before generating explanations (especially for the few-shot GPT3 examples)?**
>
> First, note that the success of prompting is dependent on how similar the task is to pre-training data [14]. For example, instead of asking "who is the president of the US?", if I ask "The president of the US is _," it becomes easier for the model because filling in the blank is much closer to the next word prediction loss function used in pre-training. Similarly, in our case of multimodal scientific reasoning, the model performance is dependent on how similar our formulated task is to the text that is seen during pre-training. In textbooks, specifically on mathematical reasoning questions, we usually have an answer field right after the question, and then we have a detailed solution that follows, not the other way around. Since the big language models are pre-trained with textbooks and internet data that follow this format of an answer and then a solution, our representation (ALE) is similar to the pre-training data and that helps the model in improving performance.
>
> [14] Pre-train, Prompt, and Predict: A Systematic Survey of Prompting Methods in Natural Language Processing, 2022
>
> Second, the pre-trained large language models learn to generalize to unseen instances via few-shot learning from a couple of in-context examples. Provided with in-context examples in the format of QCM→ALE, pre-trained GPT-3 would learn the intrinsic relationship between the input question and the final answer, and understand the corresponding lectures and explanations required for the correct answer. Therefore, GPT-3 (CoT) could benefit from in-context examples to improve the reasoning ability, thus leading to better performance on question answering and CoT generation on the test examples in SQA.
>
> Third, to recall, our SQA dataset features long annotations of lectures and explanations. There could be huge performance drops for formats such as QCM→LEA, where lectures and explanations appear before answers. Note that GPT-3 (CoT) addresses the SQA task as a text generation problem. If the pre-trained GPT-3 is formalized to generate the long lecture and explanation first (e.g., QCM→LEA), as the lecture size is large, often the answer at the end can be truncated. Similarly, in the case of the prediction, model can just predict the lecture and may run out of tokens (note that maximum output token size is limited by the GPT-3 API) for generating the actual answer.
>
> > **W2: Comparisons to QCM-LEA or QCM-ELA.**
>
> Thank you for pointing out the suggestion. We did have these results and have added these results in the revised paper, and we’d like to include them below as well.
> - QCM→ELA, 51.5% (added in the revised paper)
> - QCM→LEA, 55.4% (added in the revised paper)
> - QCM→ALE, 73.6% (our final model in the submitted paper)
>
> There are large performance drops for QCM→ELA and QCM→LEA because the GPT-3 model stops earlier if it generates the stopping symbol or uses up the tokens when generating the long text of lectures and annotations before arriving at the final answers.

---

### Official Review · Reviewer_hjMW · 2022-07-22

**Rating:** 7
**Confidence:** 4
**Soundness:** 3 good
**Presentation:** 4 excellent
**Contribution:** 3 good

**Summary:**

This paper provides a novel dataset for science question answering, SQA. Specifically, this paper studies a dataset for evaluating the reasoning/explanation performance of QA models. The authors claimed that existing science question answering datasets either 1) lack annotations for explanations for the answers or 2) has a small data scale/set of topics. SQA consists of 21K examples, a comparably large-scale dataset compared to existing science QA datasets. SQA covers more topics than existing datasets, and SQA is a multi-modal dataset; this dataset provides image context as well as textual context. This paper shows the validity of SQA by showing that the answer prediction performance increases when QA models are fine-tuned on SQA. Also, this paper conducts explanation prediction experiments. In this experiment, QA models predict the annotated explanations/lectures, and humans evaluate the predictions. The experimental results indicate that training QA models on SQA increases the explanation prediction (reasoning) performance of QA models.

**Questions:**

Q1: This paper provides SQA for better evaluation of the reasoning performance of QA models (lines 29-30). How can we know the reasoning performance of QA models with this dataset? Does predicting the correct explanation guarantee that the model’s prediction (answer) is based on the correct reasoning processes? Although some dataset papers in this field claim similar arguments (evaluating the reasoning performance of QA models can be done by explanation generation), the explanation prediction task is insufficient to argue that SQA enables evaluating QA models’ reasoning performance.

Q2: Line 58-59: How much improvement can we make with other science QA datasets? Please compare SQA with other datasets to show SQA's efficacy. One possible experiment is fine-tuning QA models on the SQA and TQA datasets, then comparing how much improvement the QA models get.

Q3: The image captioning model also affects the models’ performance. How can you conclude that current VQA models are not generalized to process the challenging questions in SQA (lines 229-230)? Would it be possible that the poor performance came from the poor performance of the image captioning model used in this paper?

Q4: In Table 4, the authors reported the semantic similarity scores computed by Sentence-BERT. However, reporting the absolute similarity scores is not a standard way to show semantic similarity. In semantic textual similarity, ranking-based evaluation metrics are usually used. Please use rank-based metrics such as top-k, recall@K, and MRR.

**Minor questions from here**

Q5: What is the reason for choosing social science and language science?

Q6: What is the difference between “Unique” questions and “Total” questions (line 133)?

Q7: In Table 4, what is the reason for using the BLEU-1 score instead of BLEU-4, a more widely used evaluation metric?

**Limitations:**

The authors have addressed the limitations and potential negative social impact of their work.

**Strengths And Weaknesses:**

Strengths

1. The proposed dataset, SQA, is a large-scale dataset and covers more diverse science topics. Also, SQA consists of more features such as detailed lectures and explanations related to the given question (line 129).
2. This paper showed that predicting explanation increases the overall SQA performance (Table 3).
3. This paper is well written and easy to follow.
4. This paper provides a sufficient analysis of SQA.

Weaknesses

1. This paper claims that SQA is a better dataset for evaluating the reasoning performance of QA models. However, this paper does not compare SQA and other science QA datasets (see Q2 in the “Questions” section for more details).
2. I have left my concerns in the “Questions” section.

---

> ### Author Response · Authors · 2022-08-02
> **Responses to Reviewer hjMW (R2) (Cont.)**
>
> > **Q4: Evaluation metrics for the generated lectures and explanations.**
>
> Thanks for your suggestion of using rank-based metrics. Recently, BERT and its variants have been widely used in tasks of sentence-pair regression and semantic textual similarity [7,8,9,10,11]. These models are capable of computing the similarity score of two sentences in terms of semantic meaning instead of extract match or n-gram match. Thus, we report the similarity score of the generated explanations and annotated explanations using the Sentence-BERT model [8].
>
> We also report metrics like BLEU-1/4, ROUGE-L, and human evaluation scores because they are widely used in text generation tasks like machine translation, question answering, and image captioning. Rank-based metrics such as top-k might have some limitations in our work because the explanations are generated from sequence-to-sequence models instead of ranking explanation candidates. As there is no exact match between generated explanations and annotated explanations, the top-k scores for different baselines are 0. However, it is still possible to transfer our similarity score to rank. We would appreciate it if you could clarify your suggestion a bit further, and we would love to add this metric to our next version.
>
> [7] SemEval-2017 Task 1: Semantic Textual Similarity Multilingual and Crosslingual Focused Evaluation, 2017
>
> [8] Sentence-BERT: Sentence Embeddings using Siamese BERT-Networks, 2019
>
> [9] RoBERTa: A Robustly Optimized BERT Pretraining Approach, 2019
>
> [10] XLNet: Generalized Autoregressive Pretraining for Language Understanding.
>
> [11] SimCSE: Simple Contrastive Learning of Sentence Embeddings, 2022
>
>
> > **Q5: What is the reason for choosing social science and language science?**
>
> Inspired by the definition of science on Wikipedia [12], there are several branches of science: natural science, social science, formal science, and so on. Besides, many questions of language science overlap with domains in the social sciences, life sciences, culture, and humanities, thus making language science a multidisciplinary field [13]. We believe that incorporating questions of social science and language science into the proposed SQA dataset makes SQA a more reasonable dataset to diagnose the multi-modal reasoning abilities of existing models over a wide range of topics.
>
> [12] Science, https://en.wikipedia.org/wiki/Science
>
> [13] The Science of Linguistics, https://www.linguisticsociety.org/resource/science-linguistics
>
> > **Q6: What is the difference between “Unique” questions and “Total” questions (line 133)?**
>
> "Unique" questions refer to different questions where we only count one for duplicate questions, while total questions are the whole number of questions in the dataset. Normally, the number of unique questions is smaller than the total number of questions. Note that examples that share the same questions appear with different contexts, choices, and answers in the SQA dataset. As shown in the table, there are 21,208 questions "in total", and 9,122 unique questions. We have clarified it in the revised paper.
>
> > **Q7: In Table 4, what is the reason for using the BLEU-1 score instead of BLEU-4, a more widely used evaluation metric?**
>
> We have the result scores of BLEU-4 and added them in the revision:
> - UnifiedQA_BASE (CoT), QCM→ALE: 0.370
> - GPT-3 (CoT), QCM→AE: 0.048
> - GPT-3 (CoT), QCM→ALE: 0.052
>
> However, these scores of BLEU-4 are inconsistent with those of other automatic metrics like BLEU-1, ROUGE-L, and (semantic) Similarity. The reason is mainly that UnifiedQA_BASE (CoT) is fine-tuned on the SQA dataset and it tends to learn similar N-gram patterns in the training data.  UnifiedQA_BASE (CoT) can generate "plausible" lectures and explanations that are similar to the annotations in the training data. Thus, we can find that the score for UnifiedQA_BASE (CoT) is unusually higher than that for GPT-3 (CoT). Therefore, BLEU-4 might not be an ideal evaluation metric for our setup.

---

> ### Author Response · Authors · 2022-08-02
> **Responses to Reviewer hjMW (R2)**
>
> Dear reviewer, we really appreciate your effort in reviewing our paper, and thank you for your helpful comments. We are glad that you recognize that our dataset is novel, large-scale, and covers diverse science topics. We are encouraged to see that you note our designed models boost the overall SQA performance via CoT. We address your questions below:
>
> > **Q1: How can we know the reasoning performance of QA models with this dataset?  Does predicting the correct explanation guarantee that the model’s prediction (answer) is based on the correct reasoning processes?**
>
> It is really a good question! As discussed in Table 1 and Line 122-129, SQA is a new large-scale multi-modal question answering dataset with multi-modal contexts across a wide range of scientific domains. To answer the SQA questions well, QA models are required to have the abilities of joint reasoning of multi-modal inputs, commonsense reasoning and knowledge acquisition for domain-specific topics, and multi-hop reasoning. Besides, as SQA contains the detailed annotations of lectures and explanations, it serves as an important benchmark to diagnose the models’ ability to provide the evidence to reveal the reasoning steps to arrive at the answers.
>
> The consistency and coherence of large pre-trained language models are still understudied and some recent work [6] has been proposed to investigate these issues. The failure cases visualized in Figures 16 and 17 in the appendix show that GPT-3 (CoT) could predict the correct answers with wrong explanations or generate the wrong answers with gold explanations.
>
> [6] Are NLP Models really able to Solve Simple Math Word Problems? 2021
>
> > **W1/Q2: Comparison of SQA and other science QA datasets. How much improvement can we make with other science QA datasets?**
>
> Thank you for your suggestion! In this work, we construct a large-scale science QA dataset with detailed annotations of lectures and explanations across diverse scientific topics. We further compared various baselines on SQA and designed models with chain-of-thought prompting in both few-shot and fine-tuning settings. It would be meaningful to evaluate how large language models perform on other science QA datasets.
>
> However, SQA is the only multi-modal science question answering dataset that features the lectures and explanations as evidence for the answers. We are unable to do this experiment because other multi-modal science datasets don't contain explanations. We hope SQA will encourage the development of more similar resources, and that will eventually facilitate the dataset comparison.
>
> > **Q3: Are current VQA models not generalized to process the challenging questions in SQA? Discuss reasons for the poor performance of current VQA models.**
>
> Thanks for your insightful comments! Based on the current experiments on SQA, current SOTA VQA models such as VisualBERT and ViLT don't perform well in both the textual-only contexts (the highest accuracy is 66.96%) and multi-modal contexts (the highest accuracy is 62.17%).
>
> The main reason is that current VQA models are mostly trained on VQA (e.g., VQA, GQA) and image captioning datasets (e.g., COCO). These datasets usually feature natural images as the visual context and short lengths for both the inputs and outputs. Instead, SQA contains multimodal contexts (i.e., text, images, diagrams), and input questions with a wider length distribution (i.e., the question lengths range from 3 words to 141 words). Furthermore, SQA consists of multi-modal scientific questions with multi-modal contexts, and it features a diverse range of scientific domains across 3 subjects and 379 skill categories. All of these features make it difficult for existing SOTA VQA methods. Last, as we feed the image captions as the visual context input, VQA methods might fail to answer a set of visual-critical questions due to the missing visual information. Last but not least, textual models like UnifiedQA and GPT-3 can take advantage of explanations in the few-shot or fine-tuned setting, which VQA models can't.

---

### Official Review · Reviewer_f7az · 2022-07-24

**Rating:** 6
**Confidence:** 3
**Soundness:** 3 good
**Presentation:** 4 excellent
**Contribution:** 3 good

**Summary:**

 - The paper presents a Science Question Answering (SQA) dataset, which contains evidence and explanations in multimodalities and covers a diverse category of questions.
- Authors evaluate standard baselines as well as chain of thoughts (COT) baselines on the established dataset. Empirical result shows that COT can improve the few-shot performance. Due to the introduction of lectures and explanations annotation in SQA, it is also possible to evaluate COT under the supervised finetuning setting. And it could be shown that COT also helps.
- Human performance is measured as the upper bound of the task.

**Questions:**

- What is the motivation to concatenate evidence and explanation after the answer instead of before it?
- What is the insight that few-shot of GPT-3 is better than finetuned UnifiedQA? It is surprising that a few-shot model outperforms a supervised one, so more analysis is expected.
- For the questions without any context, how were they handled in the evaluation?

**Limitations:**

The authors adequately addressed the limitations and potential negative societal impact

**Strengths And Weaknesses:**

Strengths:

- The paper introduces a novel dataset that fills in the gap of the existing datasets as shown in table 1. Lectures and explanations annotation are introduced to facilitate future research on COT.
- Empirical of SOTA models with and without COT is evaluated to set baseline performance of this dataset
- The paper is clearly written and easy to follow

Weakness:

- See the question section

---

> ### Author Response · Authors · 2022-08-02
> **Responses to Reviewer f7az (R1)**
>
> Dear reviewer, thank you for the constructive comments and we appreciate your time and effort.  We are glad that you recognize that our dataset is 1) novel and diverse, 2) could fill in the gap in the existing datasets, and 3) facilitate future research on CoT. Furthermore, it is helpful that you recognize our experimental findings in the few-shot learning setting and that our paper is clearly written. We address your questions below:
>
> > **Q1: What is the motivation to concatenate evidence and explanation after the answer instead of before it?**
>
> The short answer to choosing the format of QCM→ALE in our final model is that it performs best for both question answering and explanation generation according to extensive experiments, as supported by the numbers in our submitted/revised paper.
>
> We hypothesize that generating the answer first before generating the more general lecture and explanation helps GPT-3 in the CoT setting to generalize better to unseen examples. Empirically, we find this to be true and find an increase in performance on 1,000 test examples when the answer is generated before the evidence (lecture, explanation):
> - QCM→EA, 56.0% vs QCM→AE, 67.6%
> - QCM→LA, 60.6% vs QCM→AL, 73.0%
> - QCM→ELA, 51.5% vs QCM→AEL, 73.5%
> - QCM→LEA, 55.4% vs QCM→ALE, 73.5%
>
> To recall, our SQA dataset features long annotations of lectures and explanations. In our final model (QCM→ALE), we prompt GPT-3 with CoT and the output generated consists of the answer, lecture, and explanation, in that order. We do this for the following reasons:
>
> - There is an implementation level hurdle in generating the answer after the evidence. As we only have access to GPT-3 via the OpenAI API, there is a chance that GPT-3 stops generating or uses up maximum tokens while generating the evidence before it can produce the answer. This leads to an obvious performance drop in the answer accuracy before obtaining the answer prediction.
>
>
> - Note that the success of prompting is dependent on how similar the task is to pre-training data [2]. For example, instead of asking "who is the president of the US?", if I ask "The president of the US is _," it becomes easier for the model because filling in the blank is much closer to the next word prediction loss function used in pre-training. Similarly, in our case of multimodal scientific reasoning, the model performance is dependent on how similar our formulated task is to the text that is seen during pre-training. In textbooks, specifically on mathematical reasoning questions, we usually have an answer field right after the question, and then we have a detailed solution that follows, not the other way around. Since the big language models are pre-trained with textbooks and internet data that follow this format of an answer and then a solution, our representation (ALE) is similar to the pre-training data and that helps the model in improving performance.
>
> [2] Pre-train, Prompt, and Predict: A Systematic Survey of Prompting Methods in Natural Language Processing, 2022
>
> > **Q2: What is the insight that few-shot GPT-3 is better than finetuned UnifiedQA?**
>
> GPT-3 has 175 billion parameters, almost 1,000 times more than the UnifiedQA model (200 million parameters for the version we used in the paper). Recent studies have shown that the performance of large pre-trained increases linearly with the increase of the logarithmic model size [3,4].
>
> Besides, GPT-3 is trained with about 500 billion tokens for a wide range of downstream language tasks, including question answering and long text generation. Also note that OpenAI's default engine for GPT-3 has recently been changed to a more powerful version, *InstructGPT*, which is a fine-tuned version of GPT3 on many datasets [5]. Instead, UnifiedQA is trained on eight datasets specifically for question answering tasks. Thus, a few-shot GPT-3 has advantages over most existing fine-tuned models, including UnifiedQA.
>
> [3] Language Models are Few-Shot Learners, 2020
>
> [4] OPT: Open Pre-trained Transformer Language Models, 2022
>
> [5] Aligning Language Models to Follow Instructions, https://openai.com/blog/instruction-following/
>
> > **Q3: For the questions without any context, how were they handled in the evaluation?**
>
> As discussed in Section 4.1 (Line 170-171, Line 178-182), the textual elements of the input are concatenated and then fed to baseline models. For example, the format of QCM→A takes the concatenation of tokens of the question text (Q), the context text (C), and multiple options (M) as inputs.
>
> For questions without any context, the context text is replaced with an empty string (added in Line 578-579 in the Appendix). The evaluation results of various baselines over different classes are presented in Table 3. The column NO is added in the revised paper (Table 3), showing the accuracy of different baselines for questions without any context.

---

### Author Response · Authors · 2022-08-02
**General responses to all reviewers**

We would like to thank the reviewers for providing us with thoughtful comments and constructive feedback.

We are encouraged that our constructed **dataset** SQA is new/novel (R1, R2, R3, R4), large-scale (R2, R5), promising (R3), and interesting (R4), and it features diverse topics/questions/domains (R1, R2, R4, R5).

We are pleased with our **experimental evaluations** on the SQA dataset, which is extensive, reasonable, and illuminating (R4, R5); our designed **method**, which has the validity of increasing answer prediction performance in both few-shot and fine-tuning contexts (R2, R3); and our **paper**, which is clearly and well written and easy to follow (R1, R2, R4, R5).

We appreciate that R1, R4, and R5 recognize that, due to the nature of including lectures and explanations, our proposed **dataset** can fill gaps in the existing datasets and further facilitate the development of future research on model reasoning.

We have incorporated the feedback and highlighted the updates in blue in the revised paper. We address the general concerns below.

**(1) Novelties and contributions of our work.**

First, to fill the gaps in existing datasets in the scientific domain, we built Science Question Answering (SQA), a new dataset containing 21,208 multimodal science questions with rich domain diversity. To the best of our knowledge, SQA is the first large-scale multi-modal science question answering dataset that features detailed lectures and explanations.

Second, the chain-of-thought reasoning in language models is still understudied after Wei et al. and Wang et al. first found that the chain of thought has the potential to improve the performance of complex QA tasks like math word problem-solving in 2022. In this paper, we extensively explore CoT prompting on SQA and show that CoT benefits large language models in both few-shot and fine-tuning learning by improving model performance and reliability via generating explanations. To be more specific,

1) CoT has been shown to be helpful in a few-shot setting (Wei et al.). We extend CoT to the fine-tuning setting and show that it improves the performance consistently.

2) CoT in the original paper [1] was shown to improve performance in the task of question answering. We show that, along with this performance, it also helps generate reliable explanations for the answer.

3) The original CoT paper [1] is in the text-only domain. We show that CoT and its variants also work in the multi-modal setting, provided we extract the visual information in natural language from the input images.

4) We also see evidence that CoT in our final setup (QCM→ALE) also helps the model learn from fewer data. This is analogous to how humans being taught with explanations learn quickly just with a few examples.

5) We further explore the upper bound of GPT-3 and find that if the GPT-3 model is fed with the gold lecture and explanation in the inputs.

6) We also investigate the effect of various parameters: (a) prompt format, (b) number of examples, and (c) sampling methods on model performance with CoT to better understand the sensitivity and statistical significance of the performance gains.

Third, our extensive experiments across VQA and pre-trained language models show that SQA is a challenge for state-of-the-art models in the multi-modal setting. This indicates that there is significant room for future work in this direction, and SQA provides a platform to facilitate those studies.

[1] Chain of Thought Prompting Elicits Reasoning in Large Language Models, 2022

**(2) Comparisons to other prompting formats.**

We did conduct comparisons of other formats such as QCM→A, QCM→AE, QCM→ALE, QCML→A, QCME→A, and QCMLE→A. However, results show that these formats are less likely to work. That is why we omitted these exploration experiments in the submitted paper to save space on the main page.

We have added these results in the revised paper, and we’d like to include them below.

- QCM→EA, 56.0% (added)
- QCM→LA, 60.6% (added)
- QCM→ELA, 51.5% (added)
- QCM→LEA, 55.4% (added)
- QCM→AEL, 73.6% (added)
- QCM→ALE, 73.6% (our final model in the submitted paper)
- QCML*→AE, 73.3% (The upper bound experiment suggested by R3)

Settings for the results above:
- Q: question, C: context, M: multi-choice options, A: answer, L: lecture, E: explanation.
- Experiments were done on 1,000 test examples.
- We adopted 2-shot learning with a sampling seed of 3.

---

### Meta-Review · Area_Chair_nDZ9 · 2022-08-29

**Recommendation:** Accept
**Confidence:** Less certain

**Metareview:**

The paper introduces a large new multimodal dataset for science question answering, and thoroughly evaluates a range of models, including a version of chain-of-thought. Reviewers agree that the paper is generally solid and well written, and the dataset is potentially useful. The major concerns around the technical novelty of the contributions, which are somewhat incremental extensions to chain of thought (e.g. with fine tuned and multimodal models). Some reviewers are also confused why generating the answer first gives better chain of thought results, because this appears inconsistent with the step-by-step reasoning explanation of chain of thought, and this point could be better explained. Overall I think the submission is borderline, leaning towards acceptance.

**Award:**

No

---

### Decision · Program_Chairs · 2022-09-14

Accept